# Position: World Models Should Be Divided into Three Layers: Physical-Chemical Evolution, Biological Evolution, and Human-Created Modifications

## Abstract

This paper argues that world models should be constructed by dividing the world's evolution into three stages: physical-chemical evolution, biological evolution, and human-created modifications. Current mainstream world models often map physical laws, biological behaviors, and human-designed rules into a single latent space, leading to confusion in causal relationships, instability in reasoning, and limited generalization capabilities. Starting from the perspective of world evolution, this paper analyzes the fundamental differences in evolutionary mechanisms, causal structures, and external manifestations between physical systems, biological systems, and human society systems. It argues that layering these levels is a necessary prerequisite for building reliable world models. Finally, the paper calls for the adoption of layered construction as a core method in world model research, to support more interpretable and controllable intelligent systems' understanding of the world.

## 1. Introduction: From "Looking Like the World" to "Understanding the World"

World models are widely regarded as an important cornerstone of general artificial intelligence, with the goal of constructing a predictable, inferable, and decision-usable environmental representation internally. From Dreamer to recent large-scale video generation and VLA models, mainstream methods mostly follow the same path: learning the dynamic structure of the world through high-dimensional perceptual data.

This paradigm is engineeringly successful, but it implies

---
[1]Anonymous Institution, Anonymous City, Anonymous Region, Anonymous Country. Correspondence to: Anonymous Author <anon.email@domain.com>.

Preliminary work. Under review by the International Conference on Machine Learning (ICML). Do not distribute.

a key assumption: the world can be compressed into a unified, continuous latent representation space. Cognitively, this assumption is analogous to the early stage of human infants understanding the world through sensory experience — relying on apparent correlations rather than mechanistic understanding.

The problem lies in the fact that the maturity of human intelligence stems not just from "seeing more," but from gradually learning to distinguish the causal sources of different tiers:

- Why do apples necessarily fall (physical laws),
- Why do animals exhibit goal-directed behaviors (biological adaptation),
- Why does a red light mean stopping (human agreement).

Current world models precisely lack this ability to distinguish. This paper argues that this problem is not due to insufficient data or model scale, but because world models violate the evolutionary ontology of the world at the structural level.

## 2. The Three-Tier Intrinsic Structure of World Evolution

From the perspective of cosmic evolutionary history, the world is not shaped by a single mechanism, but has undergone three essential "phase transitions," each introducing new causal mechanisms and emergent properties.

### 2.1. Physical-Chemical Evolutionary Layer: A Purpose-Free Necessity World

The physical-chemical evolutionary layer forms the fundamental basis of the world, starting with the Big Bang and covering the formation processes from elementary particles, atoms, and molecules to galaxies, planets, and geological systems.

The core characteristics of this layer are:

- **Causal mechanism:** Strictly constrained by physical and chemical laws, with high determinism;

- **Time scale:** From nanosecond-scale physical processes to geological epochs;
- **External performance:** Continuous, differentiable, and purpose-free.

In this layer, there is no "intention" or "adaptation," only necessity and constraint. For world models, this layer provides unbreakable hard constraints rather than empirical rules that need to be "guessed" through data.

### 2.2. Biological Evolutionary Layer: A World of Adaptation and Selection

Under specific physical and chemical conditions, living systems emerged, and the world entered the second evolutionary stage. Biological evolution introduced a new causal logic: systems began to adjust their structures around "survival and reproduction."

The key characteristics of this layer include:

- **Causal mechanism:** Evolutionary dynamics of variation-selection-inheritance;
- **Time scale:** Medium to long-term evolution based on generations;
- **External performance:** Goal-directed behaviors, adaptive strategies, and ecological structures.

Biological systems no longer merely passively obey physical laws, but improve their survival probability by changing their own structures and even the environment. This "adaptive causality" cannot be directly derived from the pure physical layer.

### 2.3. Human-Modified Layer: A World of Symbols, Rules, and Design

The emergence of human civilization marks the beginning of the third evolutionary layer. The core of this layer lies not in natural selection, but in conscious design and social cooperation.

Its characteristics are manifested as:

- **Causal mechanism:** Intention-driven, symbolic systems, and institutional constraints;
- **Time scale:** From immediate decision-making to civilizational evolution;
- **External performance:** Tools, buildings, language, culture, and software systems.

This layer is not a simple continuation of natural evolution, but a systematic rewriting of the world by humans through design languages and social rules.

## 3. A Three-Tier Decomposition of the World: Why Flat World Models Fail

Current world models—whether vision-centric simulators, latent dynamics models, or end-to-end predictive architectures—implicitly assume that the world is representable within a single homogeneous modeling space. This assumption is the fundamental reason they struggle to scale beyond narrow environments.

We argue that the objective world is structurally stratified, not merely complex. Its complexity does not arise from scale alone, but from the coexistence of heterogeneous generative principles that obey fundamentally different laws. Treating these principles as if they were learnable within a unified latent space leads to three systemic failures: spurious correlations, brittle generalization, and non-interpretable reasoning.

### 3.1. The Physical Tier: Unlearnable Constraints and Invariant Laws

The physical world is governed by conservation laws, geometric constraints, and causal regularities that are not optional and not learnable from data alone in the general case. Gravity, thermodynamics, rigid-body constraints, and signal propagation limits exist independently of observation frequency or dataset coverage.

End-to-end learning systems attempt to approximate these laws statistically, but such approximation:

- Requires prohibitive data coverage,
- Fails catastrophically outside the training distribution,
- And conflates physical necessity with empirical regularity.

From a modeling perspective, the physical tier should be treated as a domain of hard constraints and invariants, where learning is secondary to constraint enforcement. This tier defines what is possible, not what is likely.

### 3.2. The Biological Tier: Adaptive Dynamics Under Physical Constraints

Above the physical tier lies the biological world: systems that adapt, optimize, and evolve, but always under physical constraints. Biological agents do not violate physics, yet their behavior cannot be reduced to physics alone.

This tier is characterized by:

- Selection pressure and fitness landscapes,
- Population-level dynamics and emergence,
- Heuristic, satisficing strategies rather than optimal control.

Unlike physical laws, biological rules are learnable but non-

stationary. They change over evolutionary time and across environments. Modeling them as static transition functions—as many RL-based world models do—misses the core mechanism of adaptation.

### 3.3. The Human Tier: Normative, Symbolic, and Institutional Dynamics

Human society introduces a qualitatively different layer of causality:

- Norms, laws, and institutions,
- Symbolic reasoning and language,
- Explicit goals, values, and abstractions.

Crucially, human rules are not reducible to biological fitness or physical efficiency. Social conventions may persist even when they are biologically neutral or physically inefficient.

This tier is inherently symbolic and normative. Attempting to encode it purely through sub-symbolic representations leads to opaque reasoning and value misalignment.

### 3.4. Why Flat World Models Inevitably Collapse Tiers

A flat world model forces:

- Physical constraints to be "rediscovered" statistically,
- Biological adaptation to be confused with noise,
- Human rules to be treated as soft correlations.

The result is a system that appears powerful in closed benchmarks but lacks any stable notion of causality, responsibility, or abstraction when deployed in open environments.

## 4. How Hierarchical Architecture Changes the Way AI Learns the World

### 4.1. From End-to-End Fitting to Structured Understanding

A hierarchical world model explicitly separates what must be obeyed, what can be adapted, and what should be reasoned about.

Instead of forcing a single model to simultaneously learn:

- Conservation laws,
- Evolutionary strategies,
- Social norms,

the hierarchy allows each tier to specialize in modeling phenomena appropriate to its nature.

This distinction enables AI systems to recognize unlearnable constraints (e.g., physics), learnable regularities (e.g., behavior patterns), and negotiable rules (e.g., social conventions), rather than blending them into a single opaque latent space.

### 4.2. Natural Emergence of Interpretable Reasoning Paths

In a hierarchical architecture, decision-making naturally decomposes into tier-specific justifications:

- "This action is forbidden due to physical safety constraints."
- "This strategy is suboptimal given biological survival dynamics."
- "This behavior violates human norms or institutional rules."

Such interpretability is structural, not post-hoc. Unlike saliency maps or probing methods applied to unified models, hierarchical interpretability is intrinsic to the architecture itself.

This property is critical for:

- Safety-critical systems,
- Human-AI collaboration,
- Legal and ethical accountability.

### 4.3. Modular Generalization as a First-Class Capability

When environmental changes affect only one tier, hierarchical models allow local adaptation:

- A new physical environment updates only the physical layer,
- A change in population dynamics updates the biological layer,
- A policy shift updates the human layer.

In contrast, end-to-end models require global retraining, leading to catastrophic forgetting or unintended side effects.

This modularity transforms generalization from an emergent accident into a designed capability.

## 5. Inter-Tier Coupling: The True Source of World Complexity

While the tiers are distinct, the world's richness emerges from their bidirectional coupling.

### 5.1. Upward Causality: Constraints Shape Possibilities

Physical constraints define feasible biological adaptations; biological capacities delimit human institutions. For example:

- Energy constraints limit evolutionary strategies,
- Cognitive limitations shape social norms and legal systems.

Ignoring upward causality leads to models that propose impossible or unsafe actions.

## 5.2. Downward Causality: Higher Tiers Reshape Lower Ones

Human activity fundamentally alters both biological and physical environments:

- Urbanization reshapes ecosystems,
- Industrialization alters climate dynamics,
- Medical intervention redirects evolutionary trajectories.

A world model that lacks explicit downward causality can only perform shallow prediction, not genuine understanding.

## 5.3. Why Cross-Tier Feedback Must Be Explicitly Modeled

Implicitly mixing tiers in latent spaces obscures causal directionality. Explicit inter-tier interfaces allow:

- Controlled information flow,
- Causal attribution,
- Counterfactual reasoning across abstraction levels.

Without this, world models remain simulators, not reasoning agents.

## 6. A Technical Route Toward Constructing a Three-Tier World Model

### 6.1. Hierarchical Representation and Specialized Modeling

- **Physical tier:** Physics-informed neural networks, constraint solvers, and symbolic engines to enforce invariants.
- **Biological tier:** Evolutionary algorithms, population dynamics models, and agent-based simulations capturing adaptation.
- **Human tier:** Large language models, knowledge graphs, and neuro-symbolic systems for normative reasoning.

Each tier is optimized for its own epistemic structure rather than forced into a unified formalism.

### 6.2. Inter-Tier Interfaces and Causal Alignment

Rather than implicit feature sharing, tiers communicate through explicit interfaces:

- State abstractions,
- Causal signals,
- Constraint propagation mechanisms.

Causal reasoning ensures that information flows in logically valid directions, preventing semantic leakage between tiers.

### 6.3. Training Paradigm and Hierarchical Evaluation

Training should proceed in phases:

- Physical consistency verification,
- Biological adaptivity assessment,
- Human-level normative alignment.

Evaluation must be tier-aware; otherwise, systems risk "performing well at the wrong level," such as achieving social objectives by violating physical safety.

## 7. Conclusion and Call

This paper proposes and argues for a clear stance: world models must be constructed based on the three-tier structure of world evolution. This is not an improvement in engineering techniques, but a respect for the causal structure of the world.

We call for world model research to shift from "unified latent space" to "hierarchical evolutionary modeling," promoting intelligent systems from "seeming reasonable" to "being fundamentally correct."

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

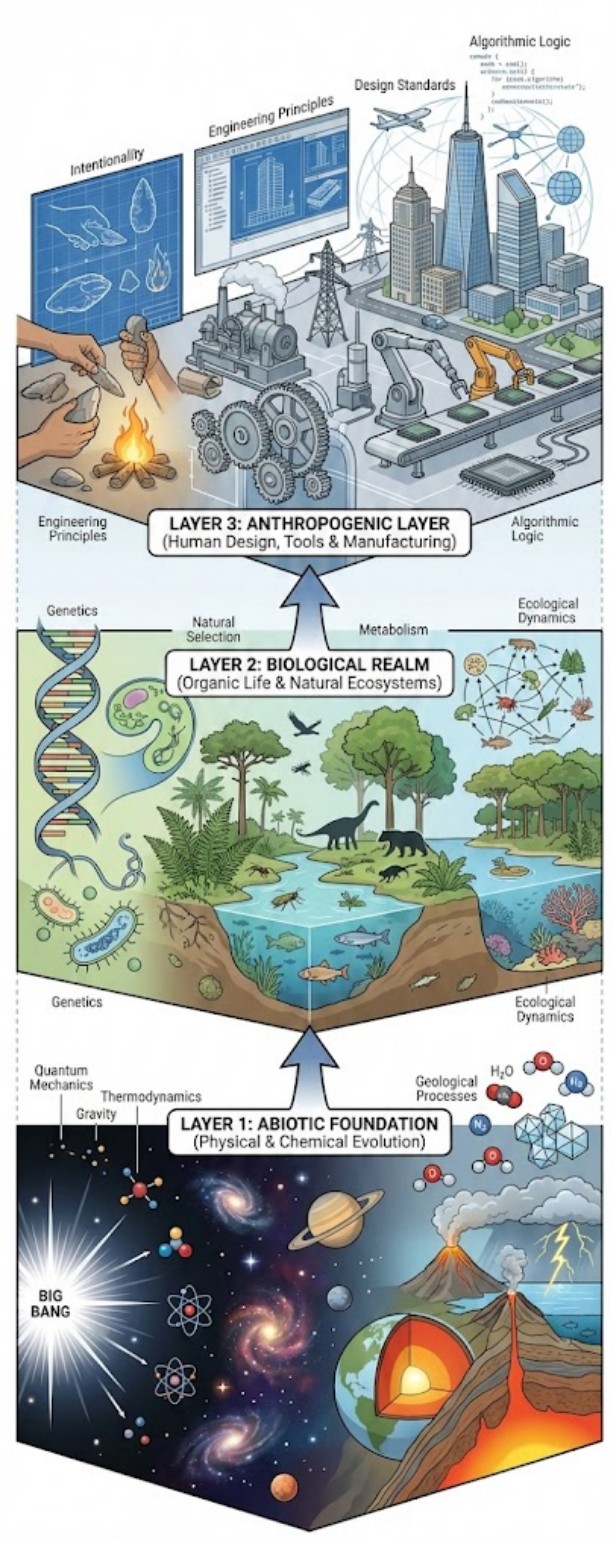

*Figure 1.* Illustration of the three-tier intrinsic structure of world evolution.

