# OpenReview forum: "Position:World Models Should Be Divided into Three Layers: Physical-Chemical Evolution, Biological Evolution, and Human-Created Modifications"
_ICML.cc/2026/Position_Paper_Track — Submitted to ICML 2026 Position Paper Track_

### Official Review · Reviewer_ugpc · 2026-02-24

**Significance:** 3
**Argument Clarity:** 3
**Rating:** 3
**Confidence:** 4

**Questions:**

Should the world model be tied to the application?
Can the world model be considered a module within a layered system solution?
It's claimed that "The physical world is governed by conservation laws, geometric constraints, and causal regularities that are not optional and not learnable from data alone in the general case". Is that true？How do animals learn physical laws/constraints? Are they also statistically "rediscovered"?
World models can be designed hierarchically, but not necessarily according to the hierarchy in this article, right? For example, the System 1 & 2 layered architecture, which draws inspiration from the brain's "System 1" (fast, intuitive) and "System 2" (slow, deliberate) thinking.
What are the advantages of a non-hierarchical (flat) structure compared to a hierarchical structure?

**Alternative Views Section:**

No

**Compliance With Llm Reviewing Policy A Conservative:**

Affirmed.

**Discussion Potential:**

3

**Paper Summary:**

This article advocates for a three-tiered approach to world modeling: physicochemical evolution, biological evolution, and human modification. It analyzes the fundamental differences in evolutionary mechanisms, causal structures, and external manifestations among physical, biological, and human social systems, and demonstrates the necessary prerequisites for constructing reliable world models. Finally, the article calls for layered construction as a core methodology in world model research to support more interpretable and controllable understandings of the world by intelligent systems.

**Position:**

Yes

**Position In Title:**

Yes

**Related Work:**

2

**Strengths And Weaknesses:**

Strengths:
1. The authors clearly articulate their position and the reasons supporting it.
2. This topic is closely related to and significant within the ICML community, and is expected to spark discussion about world models.

Weaknesses:
1. This paper lacks discussion and citation of existing work on hierarchical world modeling.
2. They do not adequately discuss why so many works adopt an "alternative perspective" to construct world models.

**Support:**

2

---

### Official Review · Reviewer_Lmjv · 2026-03-11

**Significance:** 2
**Argument Clarity:** 3
**Rating:** 3
**Confidence:** 3

**Questions:**

(1) The authors should add a discussion on why some researchers oppose layered construction.

(2) The authors should add a discussion on different layered schemes for world models and the advantages of the proposed layered approach.

(3) The authors should add a discussion on how to evaluate the performance of the three-layer world model.

**Alternative Views Section:**

No

**Compliance With Llm Reviewing Policy A Conservative:**

Affirmed.

**Discussion Potential:**

2

**Final Justification:**

This is my final decision.

**Paper Summary:**

This paper proposes a perspective on building world models, advocating for dividing the construction into three levels: physical-chemical evolution, biological evolution, and human-created modifications.

**Position:**

Yes

**Position In Title:**

Yes

**Related Work:**

2

**Strengths And Weaknesses:**

**Strengths**

(1) World models are an important research direction in the field of machine learning.

(2) The paper presents a clear viewpoint, advocating for a layered construction of world models.

(3) This perspective may spark some controversy and stimulate discussion, for example, regarding the division of world evolution levels.

**Weaknesses:**

(1) The paper focuses primarily on demonstrating the advantages of layered construction but does not discuss in detail why some researchers oppose this approach.

(2) The paper does not compare different layered schemes for world models, nor does it justify the advantages of the layered approach proposed by authors.

(3) The paper does not clearly explain how to evaluate the performance of the three-layer world model, which may lead to disagreements among researchers regarding evaluation criteria.

**Support:**

2

---

### Official Review · Reviewer_xaR5 · 2026-03-12

**Significance:** 2
**Argument Clarity:** 1
**Rating:** 2
**Confidence:** 3

**Questions:**

NA

**Alternative Views Section:**

No

**Compliance With Llm Reviewing Policy A Conservative:**

Affirmed.

**Discussion Potential:**

1

**Final Justification:**

No rebuttal provided. My comments and score remains the same.

**Paper Summary:**

This paper advocates for a position that world models should be divided into three phases: physical-chemical, biological evolution, and human created modifications. They argue that current world models lack the ability to distinguish causal sources of distinct tiers. They motivate that because our world is fundamentally composed of heterogeneous susbsystems that obey fundamentally different laws, and that they should be treated differently if world models were to have robust generalization.

**Position:**

Yes

**Position In Title:**

Yes

**Related Work:**

1

**Strengths And Weaknesses:**

Strengths:

1) The idea that current world models have issues and taking a position to work towards fixing them makes sense.


Weaknesses:

1) None of the references are refereed to in the main text, making this paper hard to follow.
2) It appears to me that the manuscript was written in a rush and a lot of the bullets dont have sufficient evidence.
3) Generally, I do not think it is a publication ready position paper.

**Support:**

2

---

### Official Review · Reviewer_g8zQ · 2026-03-13

**Significance:** 1
**Argument Clarity:** 1
**Rating:** 2
**Confidence:** 3

**Questions:**

- Why it should be these three layers, but not other categorizations?
- Is there any concretely pipeline to achieve this ambitious agenda?

**Alternative Views Section:**

No

**Compliance With Llm Reviewing Policy A Conservative:**

Affirmed.

**Discussion Potential:**

2

**Final Justification:**

No rebuttal submitted. Evaluation remains the same.

**Paper Summary:**

This position paper argues that world models need to be structured into three layers: physical-chemical layer, biological layer, and a “social norm” layer. It then describes what an ideal world model should consist of and the specific descriptions of each layer. The paper then describes the limitations of the current world models, and concludes with a technical route and a call for action.

**Position:**

Yes

**Position In Title:**

Yes

**Related Work:**

1

**Strengths And Weaknesses:**

Strengths:

- The discussed topic of world models is a timely and trendy topic. It is a popular topic at the current era, and should interest a large community in ICML.

Weaknesses:

- The proposed alternative view for world model appears to be not technically achievable from the current tech stack, yet the authors fail to provide convincing evidence that how this can be achieved realistically.
- It is not clear why it should be these three layers, but not other categorizations. For examples, why should physical phenomena and chemical reactions to be categorized to the same layer? Or why cannot we add other layers of economics or politics? The current categorization appears to be random without a insightful reasoning process.
- There is no alternative views in the main text. This violates the structural requirement of the call-for-paper for this position paper track.
- The manuscript is very poorly written. It doesn’t include any citation in its main text. Most paragraphs appear to be very sparse with very weakly structured illustration.

**Support:**

1

---

### Decision · Program_Chairs · 2026-04-30

**Decision:**

Reject

**Comment:**

The reviewers agreed that the topic of the paper is timely, addresses an important problem, and is likely to interest the ICML community.

They also raised serious concerns; therefore, the paper cannot be accepted to the conference at this time.
The reviewers noted:

* The authors failed to provide convincing evidence of how the proposed method can be realistically achieved.
* It is not clear why the listed layers should be used instead of alternative choices.
* The paper lacks discussions of existing work on hierarchical world modeling.

The authors did not address the reviewers' questions.

The AC has read the authors’ rebuttals and comments and has incorporated them into the decision-making process.